# GenCAD: Image-Conditioned Computer-Aided Design Generation with Transformer-Based Contrastive Representation and Diffusion Priors

**Md Ferdous Alam**  *mfalam@mit.edu*
*Department of Mechanical Engineering*
*Massachusetts Institute of Technology*

**Faez Ahmed**  *faez@mit.edu*
*Department of Mechanical Engineering*
*Massachusetts Institute of Technology*

**Reviewed on OpenReview:** *https://openreview.net/forum?id=e817c1wEZ6*

## Abstract

The creation of manufacturable and editable 3D shapes through Computer-Aided Design (CAD) remains a highly manual and time-consuming task, hampered by the complex topology of boundary representations of 3D solids and unintuitive design tools. While most work in the 3D shape generation literature focuses on representations like meshes, voxels, or point clouds, practical engineering applications demand the modifiability and manufacturability of CAD models and the ability for multi-modal conditional CAD model generation. This paper introduces GenCAD, a generative model that employs autoregressive transformers with a contrastive learning framework and latent diffusion models to transform image inputs into parametric CAD command sequences, resulting in editable 3D shape representations. Extensive evaluations demonstrate that GenCAD significantly outperforms existing state-of-the-art methods in terms of the unconditional and conditional generations of CAD models. Additionally, the contrastive learning framework of GenCAD facilitates the retrieval of CAD models using image queries from large CAD databases, which is a critical challenge within the CAD community. Our results provide a significant step forward in highlighting the potential of generative models to expedite the entire design-to-production pipeline and seamlessly integrate different design modalities.

## 1 INTRODUCTION

Generating 3D shapes is a fundamental part of the engineering design process that morphs the design from creative intuition to digital 3D shapes. Professional engineers use modern Computer-Aided Design (CAD) models to digitally represent 3D shapes for applications in automotive, aerospace, manufacturing, medical devices, consumer products, and architectural designs. Engineering design of such 3D solid modeling is a complex sequential process that requires human expertise and intuition. Since the invention of CAD, most CAD modeling tasks have been manual and typically go through a long iterative process to finalize a design with desired requirements. Additionally, modern CAD software is highly non-intuitive for humans and comes with a steep learning curve for most practitioners (Piegl, 2005). Based on the motivations of recent literature, we argue that some portion of the 3D CAD modeling process can be automated using learning-based approaches, which can offer tremendous acceleration in the design and manufacturing pipeline. An important first step in this direction is to build generative models that can provide valid and real-world 3D CAD models based on user intuitions. As most of the generative model architectures are proposed for image or text generation, a carefully designed framework is needed for 3D CAD modeling in engineering computational design tasks.

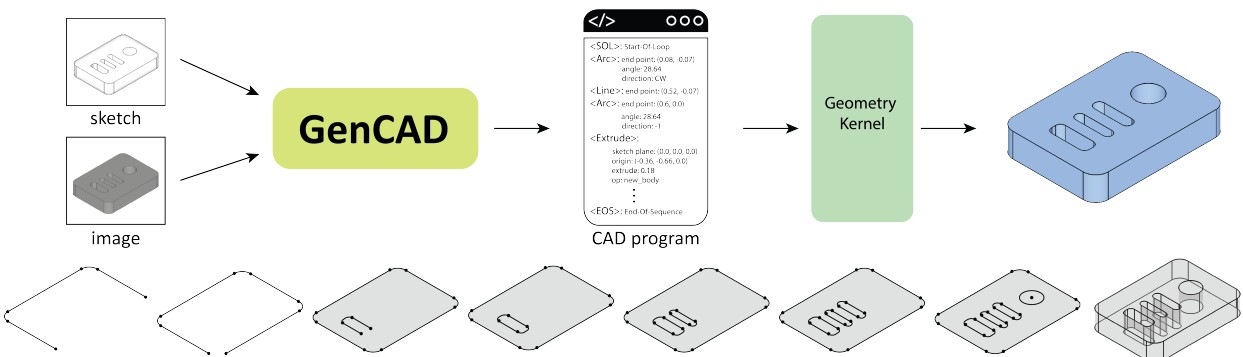

Figure 1: GenCAD demonstrates generative CAD models conditioned on images or sketches, where the CAD model is generated sequentially using a language-like representation of CAD operations

Generative models, built upon large neural network architectures, have recently shown impressive performance in image and text generation tasks (Ramesh et al., 2021; Razavi et al., 2019; Saharia et al., 2022; Karras et al., 2019; Rombach et al., 2022). To align the outputs of these generative models with user intentions, they are often conditioned on additional image or text input. Inspired by the success of text and image modalities, many recent studies have focused on building generative models for 3D representations such as meshes (Geuzaine & Remacle, 2009; Feng et al., 2019), voxels (Wang et al., 2017; Alam et al., 2025), point clouds (Qi et al., 2017; Qin et al., 2019), and implicit representations (Jun & Nichol, 2023). These applications do not include CAD modeling tasks that require *editable and manufacturable representation* of 3D solids. One of the fundamental challenges in building generative models for CAD is the underlying data structure of the 3D solid model. Since the 1980s, the industry standard in Computer-Aided Design (CAD) has been the use of boundary representations (B-rep) (Weiler, 1986) that encodes geometry information in the parametric representation of surfaces, edges, and vertices. Although B-rep is more complex than other types of 3D shape representations, it offers several benefits. For example, the boundary representation of the 3D solid model is not resolution-dependent and memory-intensive such as meshes, voxels, or point clouds while also being easier to render, unlike their counterparts, such as implicit representations. Additionally, the existing off-the-shelf geometry kernels can seamlessly convert a parametric CAD command sequence or a CAD program[1] into a B-rep, which makes this representation the de facto industry standard. However, due to the complex topological relationship between geometric entities, B-rep data structure is not directly suitable as input for neural network architectures, and an intermediate network-friendly representation must be utilized. Few recent studies have developed CAD datasets that are neural network friendly, which makes it feasible to build powerful generative models for directly generating B-reps or CAD models (Koch et al., 2019; Wu et al., 2021; Willis et al., 2021; 2022). Most of these approaches utilize one of the following: graph-based representation of 3D solid models (Jayaraman et al., 2021; Lambourne et al., 2021), a sequential representation of CAD modeling operations (Xu et al., 2022; Wu et al., 2021), or a specialized variant of the B-rep such as indexed list data structure (Jayaraman et al., 2022).

Learning to generate the CAD modeling sequence can provide more usefulness than directly learning to generate a B-rep. We argue that direct B-rep generation is less attractive as it does not encode the underlying design history. The sequence of solid modeling operations, or CAD program, is critical to modern CAD software and offers a more flexible and interpretable representation than the direct B-rep. More specifically, a B-rep model can be thought of as a sequence of parametric CAD commands that can create a 3D solid shape using an off-the-shelf geometry kernel. Although this representation of 3D CAD is straightforward and scalable, most available literature focuses on the unconditional generation of CAD programs, which is not aligned with any user input. We argue that these unconditional generative models lack the true potential of generative CAD models for useful 3D solid modeling tasks. To align the generative CAD models with user intention, we propose a new type of neural network-based architecture that is conditioned on CAD images. The idea of image-to-CAD conversion or image-to-technical drawing conversion is of particular

---

[1] A sequence of parameterized CAD modeling operations is denoted as a CAD program

importance to the design community (Dori & Tombre, 1995; Dori & Wenyin, 1999; Nagasamy & Langrana, 1990; Gümeli et al., 2022; Majumdar & Seethalakshmy, 1997). Although several attempts have been made in the past, most of the approaches use heuristics and are application-specific, which makes them difficult to generalize. Our approach is largely motivated by the recent success of text-to-image and image-to-mesh literature (Alliegro et al., 2023; Tyszkiewicz et al., 2023; Xu et al., 2024a).

Here we focus on the problem of learning to generate CAD programs in terms of parametric command sequences based on CAD images. We propose a new generative model architecture, generative CAD (Gen-CAD), that can sequentially generate an entire CAD program aligned with an input CAD-image as shown in Figure 2. Our main idea is to learn the joint distribution of the latent representation of the CAD command sequences and the CAD images or sketches using a contrastive learning-based approach. Next, we develop a conditional latent diffusion model as a prior network that can generate CAD latent conditioned on the input image latent. Finally, we present a transformer-based decoder model that can create the CAD program from the input CAD latent. Note that the final output of GenCAD is not merely a 3D solid model but a CAD program, which is an entire sequence of parameterized CAD commands. This is critical because the CAD command sequence can be converted to B-rep models or other convenient representations such as mesh, point clouds, or voxels using any off-the-shelf geometry kernel. This trivial conversion allows GenCAD to be directly useful in generating conceptual 3D engineering design which has the potential to automate many complex 3D solid modeling tasks such as the reverse engineering of CAD models from image or sketch inputs (Thompson et al., 1999). Another major benefit of GenCAD is its ability to successfully perform image-based CAD program retrieval tasks, a major challenge in the engineering design community. We show that our framework can utilize the learned joint representation of CAD images and CAD command sequences to retrieve CAD programs using image input. Our contributions can be summarized as follows:

- We develop a transformer-based autoregressive model for representation learning of CAD sequences and show that our autoregressive model is more accurate in reconstructing CAD sequences than state-of-the-art models

- We propose and develop GenCAD, an image and sketch conditional generative model for CAD, and show that conditional generation of CAD outperforms unconditional models in terms of diversity, fidelity, and statistical distance

- We show that contrastive learning enables image-based CAD retrieval, which is more than $15\times$ accurate in retrieval when compared to image-to-image search

## 2 RELATED WORK

### 2.1 CAD as a language modeling problem

The recent success of deep neural networks for language modeling tasks (Brown et al., 2020; Vaswani et al., 2017) has allowed many domains to formulate language-like problems for specific applications. Although learning design intent and developing intelligent CAD systems have been topics of interest since the 1990s (Ault, 1999; Ohsuga, 1989), treating CAD as a language modeling task is a somewhat recent approach.Ganin et al. (2021) showed that CAD can be converted to a language modeling task by considering protocol buffers (PB) for describing CAD sketch structures. The authors used PB to describe various sketch entities and constraints. Extending this idea to 3D is non-trivial due to the requirement of specific protocol buffers for 3D CAD commands. Similarly, another study by Para et al. (2021) focused on developing language-like problem formulation for CAD sketches that can handle various primitives and constraints. This approach is also limited to 2D sketches, and the scalability of the approach to 3D CAD is also non-trivial. Wu et al. (2021) also focused on converting CAD as a language modeling problem. This approach utilizes the transformer architecture for sequential 3D CAD generation and develops a simple vocabulary for CAD commands. This approach is capable of both sketch and 3D operation but only a limited number of operations. Note that the generative model proposed by Wu et al. (2021) is entirely unconditional and does not allow any user input during generation.

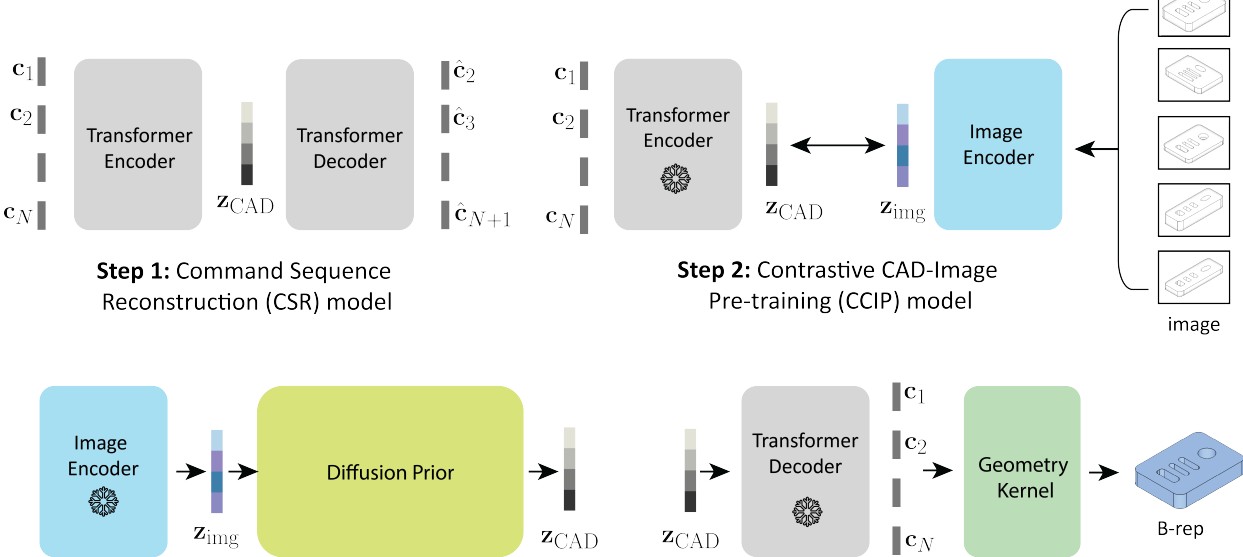

Figure 2: **GenCAD:** The proposed framework, GenCAD, consists of four steps: 1) a transformer-based encoder-decoder architecture is trained autoregressively to learn the latent representation of the vectorized CAD commands, 2) a contrastive-learning based model is used to learn the joint representation of the latent space of CAD command sequence and CAD image, 3) image conditional CAD generation can be achieved by sampling CAD latents from the diffusion model conditioned on image latents, and 4) using the trained transformer decoder to predict the CAD commands autoregressively. Note that ❄ denotes a frozen model that is not updated during training.

## 2.2 Datasets for CAD

Unlike other representations of 3D shapes such as meshes, point clouds, voxels, or implicits, there is a lack of large-scale datasets for CAD. There are only a few publicly available CAD datasets, among which the ABC dataset is the largest, containing 1 million CAD models obtained from online public repositories (Koch et al., 2019). This dataset contains B-reps but does not provide any design histories or labels. Later the DeepCAD dataset was developed by cleaning the ABC dataset (Wu et al., 2021). The DeepCAD dataset provides design history by parsing each CAD design from the Onshape public repository. In addition to this, the Fusion 360 dataset provides both design history, assembly, and face segmentation although the dataset is quite small for learning generalizable models (Willis et al., 2021; 2022; Lambourne et al., 2021). Additionally, the MFCAD and MFCAD++ datasets provide machining feature recognition data in terms of B-reps (Cao et al., 2020; Colligan et al., 2022).

## 2.3 Generative models for CAD

In addition to learning specific tasks directly from B-rep data, recent studies have also developed generative models by directly synthesizing from B-rep data structure (Xu et al., 2022; Jayaraman et al., 2022; Xu et al., 2024b). Recent generative approaches mostly focus on utilizing transformer-based architecture for generating unconditional sketches (Xu et al., 2022; Para et al., 2021) or 3D CAD (Xu et al., 2022; Wu et al., 2021; Xu et al., 2024b; Jayaraman et al., 2022). In contrast to directly synthesizing the B-rep, the CAD generation process (CAD program) is of crucial importance due to its potential in the automation of many design tasks. Most importantly, the parameterized CAD operations or commands contain valuable design history. Wu et al. (2021) developed DeepCAD, a transformer-based latent generative adversarial network (l-GAN) model, that can generate unconditional CAD. Most of the generative CAD approaches only consider unconditional generation or specific data structures such as graph representation from boundary representation of CAD

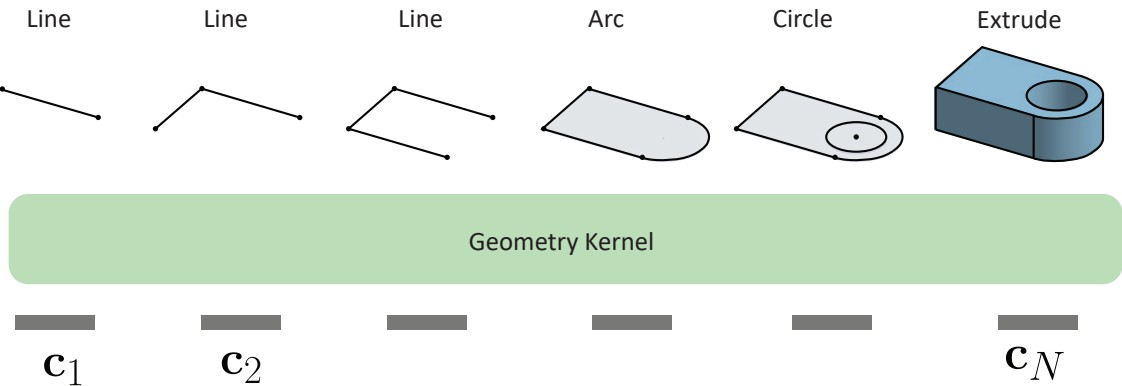

Figure 3: **CAD as a language modeling problem using geometry kernel:** Our main idea is based on the fact that real-world CAD design is sequential and a learning-based approach should capture the correlation in the sequential design from a large-scale dataset. This sequential approach converts the CAD problem into a language modeling problem where vector representation of CAD commands, $\mathbf{c}_i \in \mathbb{R}^d$, are fed into an off-the-shelf geometry modeling kernel to sequentially create the 3D solid geometry such as Boundary-representation (B-rep). In this toy example, the sequence is as follows: line $\rightarrow$ line $\rightarrow$ line $\rightarrow$ arc $\rightarrow$ circle $\rightarrow$ extrusion.

models. To align CAD model generation with user intention, a conditional input, such as an image, is needed–unlike the models developed in previous studies. To the best of our knowledge, there has been no generative model reported in the literature that can provide an entire CAD program from an image input.

## 3 METHOD

In contrast to the studies discussed in section 2, our main motivation for this work is to develop a learning-based approach that is scalable and can generate conditional 3D CAD. More specifically, we are interested in creating CAD programs that are conditioned on images. In the following, we describe our main contribution, a four-step framework, for generating image-conditional CAD models.

### 3.1 From natural language to CAD language

To represent the CAD program in a neural network-friendly representation, we opt for a natural language-like representation of a CAD program. The key idea is to represent a CAD model as a sequence of parameterized CAD commands that can be fed into a standard geometry kernel to create the 3D solid model. The goal of the parameterization is to allow the CAD command to have a fixed-dimensional vector representation. Each of the CAD commands is analogous to tokenization in natural language processing (NLP) tasks. Similarly, the fixed dimensional vector representation of the CAD command can be thought of as the embedding of a token in NLP tasks. A major difference between the tokenization of words and the tokenization of CAD commands is that the CAD commands are parameterized where the parameters ensure geometric consistency. Each CAD command represents the type of CAD operation that is being used and the associated parameters required to perform this CAD operation. An example of this mechanism is shown in Figure 3 where the vectorized CAD commands are fed into a geometry kernel to create the 3D solid geometry sequentially. Note that the type of the CAD operation is discrete and is similar to NLP tokens, but the parameter values are continuous.

Here we closely follow the CAD command representation criteria proposed by Wu et al. (2021). Specifically, each CAD command, $\mathbf{c}_i \in \mathbb{R}^{17}$, can be represented as $\mathbf{c}_i = (t_i, \mathbf{p}_i)$ where $t_i \in \mathbb{R}$ represents the command type and $\mathbf{p}_i \in \mathbb{R}^{16}$ represents the parameters of each command. Note that each CAD command has a different number of parameters but they can be converted to a fixed dimensional vector by concatenating additional parameters and masking the values for unused parameters. Additionally, the parameters are carefully chosen

so that they are sufficient for the geometry kernel to create the geometric entity. The types of CAD tokens used in this study are described in the following.

**Special tokens:** We define two special tokens ⟨SOL⟩ and ⟨EOS⟩ to represent the start of a loop of a sketch and end of sequence respectively. We draw motivation from the NLP literature which also defines special tokens to represent the start of a sentence or end of a sentence. These special tokens do not require any parameters and only indicate the state of the CAD geometry.

**Sketch token:** We define three tokens to represent a sketch; `Line` token, `Circle` token, and `Arc` token. In a geometry kernel, a line can be generated using only the end point coordinate, $(x, y)$, if the start point is known. As we are generating a CAD sketch sequentially, each sketch command provides us with the starting position for the next command. A circle can be generated based on the following three parameters; the center of the circle $(x, y)$ and the radius of the circle, $r$. Similarly, an arc can be represented by the four parameters: end point of the arc, $(x, y)$, sweep angle, $\alpha$ and directional flag of the arc, $f$ which represents either a clockwise or counter-clockwise arc.

**Extrusion token:** One of the most complex yet widely used operations in CAD is the extrusion command. Representing extrusion requires keeping track of a sketch plane, extrude direction, and distance. To represent the current sketch plane we use these six parameters: three for the orientation of the current sketch plane $(\theta, \phi, \gamma)$ and three for the origin of the current sketch plane $(p_x, p_y, p_z)$. Additionally, a scale parameter is used to represent the scale of the sketch profile $s$. Extrude distance can be represented by two parameters $(e_1, e_2)$ for each side of the sketch. Finally, to sequentially generate the solid, we need to either create a new solid body or join, cut, and intersect the extruded body from the existing solid body. So, we use a boolean parameter to indicate the type of operation of the body based on the extrude command. Finally, we use another parameter to represent whether it is a one-sided or two-sided extrude. As a result, the extrude command is represented using a total of 10 parameters.

## 3.2 Dataset

Here we use the DeepCAD dataset (Wu et al., 2021) to train and evaluate all of our models. This dataset is created based on the ABC dataset which is obtained from the publicly available human CAD designs from Onshape Inc. (Onshape, 2007). Note that the ABC dataset only contains the B-rep data and does not provide any design history. The DeepCAD dataset is created by parsing the design history of each CAD using the Onshape API. As CAD designs are complex and can be difficult to learn for sophisticated 3D shapes, this dataset is limited to sketch and extrude operations to make it more approachable for neural network-based models. Thus, this dataset does not contain other CAD operations, such as edge operations (fillets/chamfers), revolve, and mirror. The sketch operation is also limited to lines, circles, and arcs. After filtering out the ABC dataset based on these operations, the DeepCAD dataset contains $178,238$ CAD designs and is the largest CAD dataset with design history. Not all of these designs can render a 3D solid which makes it difficult to extract CAD images for our purpose. We filter this dataset based on 3D solid creation using an off-the-shelf geometry kernel, Open Cascade, and end up with a total dataset of $168,674$ CAD models with command sequences. Among these, we use $152,530$ for training, $8515$ for validation, and $7629$ for testing for all the models reported in this study. For CAD images, we create five different versions of each CAD model by changing the scale of the solid model in the $x, y$, and $z$ axes. Each image is grayscale and has dimensions of $1 \times 448 \times 448$. Note that some of the scaling operations do not create a valid CAD and we omit those images. We end up with $845,105$ images obtained from the CAD dataset. Similar to the CAD images, we also create a CAD sketch dataset by taking the canny edge of these images and adding Gaussian blur. The sketch dataset consists of $845,105$ CAD sketches.

## 3.3 GenCAD: conditional generation of CAD programs

Using the CAD language modeling formulation in section 3.1, we propose **Gen**erative **CAD** (GenCAD), a four-step framework for conditional CAD program generation. First, we develop an autoregressive transformer encoder-decoder model, the Command Sequence Reconstruction (CSR) model, using CAD command sequences as input to learn the latent representation of the final 3D solid geometry model. Second, we develop a Contrastive CAD-Image Pretraining (CCIP) model with ResNet-based image encoder and con-

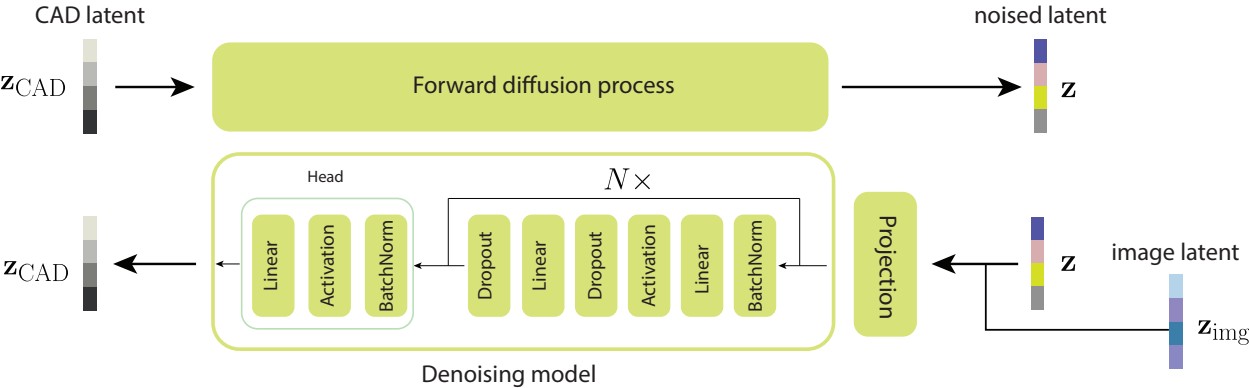

Figure 4: **CAD Diffusion Prior (CDP):** The diffusion prior takes CAD latent as well as optional image latent as input. The denoising model of the diffusion prior is based on MLP-ResNet architecture proposed by Gorishniy et al. (2021).

trastive loss to jointly learn the latent representation of the CAD image with the latent representation of the CAD command sequences from step 1. Third, we develop a CAD diffusion prior (CDP) model that can generate CAD latents, and optionally take image latents, from step 2, as input. Finally, the CAD latent samples generated from the CDP model are fed into a transformer-based decoder model that can generate the entire CAD command sequence autoregressively. To make our approach scalable for large-scale datasets and models, we utilize the pre-trained CAD encoder from step 1 during the training of the CCIP model instead of training from scratch. We also use the pre-trained CAD decoder model from step 1 for decoding the CAD latent from the CDP model. Once, we have the sequence of CAD commands, we can use any off-the-shelf geometry kernel to obtain the final 3D solid model. By combining these four steps, we show that our approach can generate valid CADs conditioned on image inputs which is shown in Figure 2.

### 3.3.1  Command Sequence Reconstruction (CSR): Latent representation learning for CAD programs

The first step of our method is to learn an underlying latent representation of the CAD program or the command sequences. This is analogous to the pre-training of modern large language models (LLM) for downstream applications. We draw motivation from the recent success of large-scale generative pre-trained models (GPT) (Brown et al., 2020; Radford et al., 2019) and propose a transformer-based autoregressive model for learning the CAD commands. The main idea is to predict the next CAD command at each timestep of the sequence. This autoregressive training is done by observing the previous CAD commands within the sequence and masking all the future commands at each timestep. We use a standard transformer encoder and decoder network (Vaswani et al., 2017) with causal masking for both the CAD sequence encoder and the decoder. As the command parameters contain discrete and continuous values, we opt for learning the distribution in the latent space instead of the parameter space.

As mentioned in section 3.1, the parameters of each CAD command can be a combination of discrete and continuous values. To unify these types of parameters, we quantize their values into 256 levels and express them using 8-bit integers similar to Wu et al. (2021). Next, we organize each command vector onto a continuous space by adding an embedding layer before the encoder network. The output of the embedding layer is a $d_z$-dimensional vector which is fed into the transformer encoder. We use $d_z = 256$ for all experiments reported in this study. The transformer encoder produces a latent representation of the CAD commands, $\mathbf{z}_{CAD,t} \in \mathbb{R}^{d_z}$ where $1 \leq t \leq N$ and $N$ is the padded sequence length. As our goal is to train a generative model using the learned representation, we create a single latent vector, $\mathbf{z}_{CAD} \in \mathbb{R}^{d_z}$, for each sequence by average pooling the sequence of latent vectors from the CAD encoder output. Constant embeddings are learned during training to project this latent vector into again a sequence of latent vectors before feeding into the decoder network. Next, the latent vector, $\mathbf{z}_{CAD}$, goes through the transformer decoder and then another embedding layer to project the continuous values into the CAD command parameters. The embedding layer

is a combination of both the CAD command embedding and the positional encoding of the sequence. Here we use the sinusoidal positional encoding from the original transformer implementation (Vaswani et al., 2017).

### 3.3.2 Contrastive CAD-Image Pre-training (CCIP): Joint representation learning of the CAD program and CAD-image

As our goal is to develop an image-conditional CAD generative model, the second step of our framework focuses on learning the joint distribution of the latent space of the CAD images and their corresponding CAD command sequences. For this purpose, we utilize the trained CAD encoder model from Step 1 to create CAD latents[2]. To generate image latents, $\mathbf{z}_{\text{img}} \in \mathbb{R}^{d_z}$, we use a ResNet-18 based architecture as the image encoder. We have experimented with larger ResNet architectures and vision transformers (ViT) but found ResNet-18 to be sufficient for our dataset (see Appendix Table 3 for ablation). First, we preprocess the input images by resizing them into $256 \times 256$ pixels, center cropping and then normalizing with $\mathcal{N}(0.5, 0.5)$. Then the images are passed through four layers with increasing dimensionality: $64, 128, 256$ and $512$. Each of these layers contains two blocks of convolutional layers. We use several dropout layers in each encoder block of the image encoder model. The output of the image encoder is $512 \times 8 \times 8$ which is projected onto the $d_z$-dimensional space using a linear layer. During training, we keep the CAD encoder frozen. Note that the training dataset contains several scaled versions of the same CAD model. This image augmentation helps the model to learn the underlying representation of similar types of CAD geometries.

### 3.3.3 CAD Diffusion Prior (CDP): Conditional generative model for CAD

One of the most critical components of our framework is the diffusion prior model that generates CAD latents conditioned on image latents. In essence, the diffusion prior model is a conditional latent denoising diffusion probabilistic model (Ho et al., 2020). The forward diffusion process is standard where the CAD latent, $\mathbf{z}_{\text{CAD}}$, is destroyed with Gaussian noise sequentially to create the noised CAD latent, $\mathbf{z}$. During the denoising process, we concatenate the noised CAD latent, $\mathbf{z}$ with image latent, $\mathbf{z}_{\text{image}}$ and use a projection layer before feeding into the denoising model. For the denoising model, we use a ResNet-MLP architecture similar to Gorishniy et al. (2021) where several blocks of ResNet-MLP blocks are used with a normalization and linear head as shown in Figure 4. For completeness, we also develop deterministic prior in addition to the diffusion prior. The deterministic prior is straightforward and deterministically predicts $\mathbf{z}_{\text{CAD}}$ from $\mathbf{z}_{\text{image}}$. For this study, we use a simple ResNet-MLP architecture as our deterministic prior model. Note that the CDP model is a generative model while the deterministic prior model only maps $\mathbf{z}_{\text{image}}$ to $\mathbf{z}_{\text{CAD}}$ deterministically. The complete architecture of the CDP model is shown in Figure 4.

### 3.3.4 CAD Decoder model: Generating CAD sequences from latent representations

Finally, we need a decoder model to convert the generated CAD latents from step 3 into CAD command sequences. To this end, we propose to utilize the pre-trained decoder part of the CSR model from Step 1 instead of training a decoder model from scratch. The use of a pre-trained decoder model is particularly useful for scaling our approach to large-scale datasets. During inference time, we would generate $\mathbf{z}_{\text{CAD}}$ from the diffusion prior model and feed it to the frozen CAD decoder model to generate the CAD command sequences $\mathbf{c}_2, \ldots, \mathbf{c}_{N+1}$. Note that the <SOL> token is prepended to this output to create the complete CAD sequence.

## 4 EXPERIMENTS

### 4.1 Training details

**Command Sequence Reconstruction (CSR) model:** For both the encoder and the decoder models, we use the standard transformer encoder and decoder architecture, respectively. Each transformer model consists of four self-attention layers and eight attention heads. The output of the final encoder layer goes

---

[2]CAD latents denote the embedding vectors obtained from encoding the CAD command sequences, whereas image latents represent the image embeddings obtained from encoding the images

through another *tanh* layer to create the latent vectors, $\mathbf{z}_{\text{CAD}}$. In total, there are 6.72 million trainable parameters in our model. To train the autoencoder model, we use the following loss:

$$\mathcal{L} = \sum_{i=1}^{N_c} \ell(\hat{t}_i, t_i) + \beta \sum_{i=1}^{N_c} \sum_{j=1}^{N_p} \ell(\hat{\mathbf{p}}_{ij}, \mathbf{p}_{ij}), \tag{1}$$

where $N_c$ is the number of CAD commands, $N_p$ is the number of parameters in each CAD command, $t_i$ is the command type and $\mathbf{p}_{ij}$ is the parameters in each command. Recall that, in this study $N_c = 6$ and $N_p = 16$ due to the modeling of the CAD sequence in section 3.1. Here we use cross-entropy loss as $\ell(\cdot, \cdot)$ in equation 1 with a weight parameter $\beta$ that regulates the relative weight of the loss of the command type and their parameters, respectively. The loss combines the CAD operation type loss and the corresponding parameter value loss.

**Contrastive CAD-Image Pre-training (CCIP) model:** For the CCIP model, we follow the standard contrastive learning procedure and use the normalized temperature-scaled cross entropy loss (Chen et al., 2020). The main idea is to maximize the similarity between a CAD image, $\mathbf{I} \in \mathbb{R}^{1 \times 448 \times 448}$ and its corresponding CAD command sequence $\{\mathbf{c}_i\}_{i=1}^N$ in a given batch of $B$ training data pairs $(\mathbf{I}, \{\mathbf{c}_i\})$. Given a batch of $B$ example pairs, we have $2B$ data points in total for the contrastive prediction. By considering the positive pair as the only positive data sample, we consider the rest of the $2(B-1)$ samples as negative data pairs. If the similarity between the data pairs is expressed as cosine similarity, $\text{sim}(\mathbf{u}, \mathbf{v}) = \mathbf{u}^T \mathbf{v}/||\mathbf{u}||\,||\mathbf{v}||$, then the contrastive loss can be defined as the following

$$\ell_{i,j} = -\log \frac{\exp(\text{sim}(\mathbf{z}_{\text{CAD},i}, \mathbf{z}_{\text{image},j})/\tau)}{\sum_{k=1}^{2B} \mathbb{1}_{[k \neq i]} \exp(\text{sim}(\mathbf{z}_{\text{CAD},i}, \mathbf{z}_{\text{image},k})/\tau)}, \tag{2}$$

where $\mathbb{1}_{[k \neq 1]} \in \{0, 1\}$ is the indicator function and $\tau$ is the temperature parameter. The final contrastive loss is then calculated as $\mathcal{L} = \frac{1}{2B} \sum_{k=1}^B [\ell(2k-1, 2k) + \ell(2k, 2k-1)]$.

For our study, we experiment with three types of image encoders: ResNet-18, esNet-35 (He et al., 2016), and ViT (Dosovitskiy et al., 2020), while keeping the CAD encoder frozen. For the ResNet-18 based CCIP model, there are 28.22 million trainable parameters in total.

**CAD Diffusion Prior (CDP) model:** Traditionally diffusion models are used for image generation tasks and thus require a 2D U-net architecture for the denoising process. In our case, the diffusion model is trained on the vector representation of the latent space. We experiment with two different denoising models to replace the U-net architecture for this purpose: an MLP-based architecture and an MLP with residual connections, ResNet-MLP. We find the ResNet-MLP model to be more effective for the denoising model and use the model for all results reported in this paper.

## 4.2 Training

The CSR model is trained using a learning rate of $1 \times 10^{-3}$ and a batch size of 512. We use the Adam optimizer with a scheduler to warm up the learning rate from zero to $1 \times 10^{-3}$ for the first 2000 steps. The CSR model is trained for 1000 epochs. The CCIP is trained using the Adam optimizer with decoupled weight decay (Loshchilov & Hutter, 2017) with the ReduceLROnPlateau scheduler and an initial learning rate of $1 \times 10^{-3}$. The CCIP model is trained for 300 epoch. For the CDP model, both the diffusion and de-noising process use 500 timesteps in our model. The CDP model is trained for 1 million timesteps with a fixed learning rate of $1 \times 10^{-5}$ where the gradient is accumulated every 2 steps. A maximum gradient norm of 1.0 is also used for training. All models are trained on an 80GB NVIDIA A100 GPU. Additional training details can be found in the Appendix.

### 4.3 Evaluation and baselines

#### 4.3.1 Command Sequence Reconstruction (CSR) model:

As our transformer-based CSR model takes the sequence of CAD commands as input, we directly compare the CSR model against the DeepCAD autoencoder (Wu et al., 2021) to show how autoregressive training helps in obtaining a more robust latent space. We mainly evaluate the reconstruction performance of each CAD command sequence in terms of three metrics as proposed by Wu et al. (2021): the mean accuracy of the CAD command generation $\mu_{\text{cmd}}$, the mean accuracy of the CAD parameter value generation $\mu_{\text{param}}$, the mean chamfer distance, $\mu_{\text{CD}}$, and the ratio of invalid shapes IR for the generated CAD against the ground truth. The mean CAD command generation accuracy and the mean command parameter generation accuracy for a set of reconstructed CAD models, $\mathcal{G}$, are defined as the following,

$$
\begin{aligned}
\mu_{\text{cmd}} &= \frac{1}{|\mathcal{G}|} \sum_{k=1}^{|\mathcal{G}|} \frac{1}{N_c} \sum_{i=1}^{N_c} \mathbb{I}[t_i = \hat{t}_i], \\
\mu_{\text{param}} &= \frac{1}{|\mathcal{G}|} \sum_{k=1}^{|\mathcal{G}|} \frac{1}{\sum_{i=1}^{N_c} \mathbb{I}[t_i = \hat{t}_i] N_p} \sum_{i=1}^{N_c} \sum_{j=1}^{N_p} |\mathbf{p}_{i,j} - \hat{\mathbf{p}}_{i,j}| < \eta \cdot \mathbb{I}[t_i = \hat{t}_i],
\end{aligned}
\tag{3}
$$

where $\mathbb{I}[\cdot]$ is the indicator function, $N_c$ is the total number of CAD commands in the sequence, $t_i$ and $\hat{t}_i$ are ground truth and predicted CAD commands respectively, $\mathbf{p}_{i,j}$ and $\hat{\mathbf{p}}_{i,j}$ are the ground truth and predicted $j$th parameter value of the $i$th CAD command respectively. As mentioned in section 3.3.1, the parameter values are quantized into 8-bit integers.

As we cannot directly calculate the chamfer distance from the B-rep data, we convert each 3D solid into a point cloud using a fixed number of points, 2000 in this case. Once we have the point clouds of both the reconstructed, $\mathcal{G}$ and the ground truth $\mathcal{S}$ shapes, we can calculate the chamfer distance, $CD$. Finally, we can calculate the mean chamfer distance as the following, $\mu_{CD} = \frac{1}{|\mathcal{G}|} \sum_{k \in \mathcal{G}} CD(k, \mathcal{S})$. We also evaluate these metrics with respect to the sequence length of the CAD commands. Intuitively, higher-length sequences should be more challenging to reconstruct due to the complexity of the CAD commands.

#### 4.3.2 Contrastive CAD-Image Pre-training (CCIP) model:

Due to the lack of existing literature on using a learning-based approach for image conditional CAD program generation, we evaluate the performance of the proposed CCIP model in terms of **image-based CAD retrieval** tasks. This is an important problem in the engineering design domain due to the wide applications of CAD retrieval from large CAD databases. The retrieval task is formulated as the following: given a CAD-image, **I**, the model should identify the CAD program, $\{c_i\}$, that aligns with this image from a set of $n_b$ CAD programs. Drawing motivation from image-based retrieval literature (Koh et al., 2024), we sample several batches of CAD examples, $n_b = 10, 128, 1024, 2048$, from the test dataset. Among these $n_b$ examples, we randomly select one sample and utilize its CAD-image as the retrieval input to the CCIP model. Next, we find the CCIP similarity between the $n_b \times n_b$ examples and pick the CAD example corresponding to the highest cosine similarity. As a baseline, we compare our methods against the image-to-image search of CAD models. For this purpose, we utilize a ResNet-18 model with pre-trained ImageNet weights.

#### 4.3.3 CAD Diffusion Prior (CDP) model:

To evaluate the image conditional generation, we compare our method against unconditional generation. For this purpose, we train an unconditional latent diffusion model from scratch as a baseline generative model. In addition to the unconditional model, we also compare the CDP model against other four baselines from literature–the latent GAN (l-GAN) model for CAD program generation Wu et al. (2021), the SkexGen model for B-rep generation (Xu et al., 2022), the BrepGen model from B-rep generation (Xu et al., 2024b), and the ContrastCAD model for CAD program generation (Jung et al., 2024). For direct comparison with the baselines, we report the metrics proposed in Achlioptas et al. (2018) for point-cloud generative models. Specifically, we create two sets of point clouds: one for ground truth shapes $\mathcal{G}$ and one for the generated

shapes $\mathcal{S}$. Next, we calculate the following: Coverage (COV) which measures the diversity of the shapes in $\mathcal{G}$, Minimum matching distance (MMD) which measures the fidelity of generated shapes, and Jensen-Shannon Divergence (JSD) which calculates the statistical distance metric between the two distributions. Details of these metrics can be found in the Appendix.

Finally, we evaluate the image-to-CAD conversion by calculating the FID score of the generated CAD programs. We create CAD programs from image inputs for the entire test data and extract their latent vectors as ground truth. Next, we create the CAD latents using conditional, unconditional, and deterministic MLP-Prior models. If the ground truth embeddings and the generated embeddings have the normal distributions $\mathcal{N}(\mu_{\mathcal{S}}, \Sigma_{\mathcal{S}})$ and $\mathcal{N}(\mu_{\mathcal{G}}, \Sigma_{\mathcal{G}})$ respectively then the FID score can be calculated as, $FID = \|\mu_{\mathcal{S}} - \mu_{\mathcal{G}}\|_2^2 + \mathrm{tr}\left(\Sigma_{\mathcal{S}} + \Sigma_{\mathcal{G}} - 2(\Sigma_{\mathcal{S}}\Sigma_{\mathcal{G}})^{1/2}\right)$. Intuitively, the FID score calculates how aligned the generated CAD programs are when compared to the test CAD-images.

## 5 RESULTS

### 5.1 CAD sequence reconstruction

We directly compare our autoregressive CSR model with the DeepCAD transformer autoencoder model. As shown in Table 1, our model outperforms the DeepCAD transformer autoencoder in every metric: mean command accuracy $\mu_{\mathrm{cmd}}$, mean parameter accuracy $\mu_{\mathrm{param}}$, mean chamfer distance $\mu_{\mathrm{CD}}$, and invalid ratio IR.

Table 1: Shape encoding performance of the proposed autoregressive encoder-decoder model. Arrows indicate whether higher or lower is better. Bold numbers represent the overall best performance.

| Method | $\mu_{\mathrm{cmd}}(\uparrow)$ | $\mu_{\mathrm{param}}(\uparrow)$ | $\mu_{\mathrm{CD}}(\downarrow)$ | IR$(\downarrow)$ |
|---|---|---|---|---|
| DeepCAD | 99.36 | 97.59 | 0.783 | 3.44 |
| GenCAD | **99.51** | **97.78** | **0.762** | **3.32** |

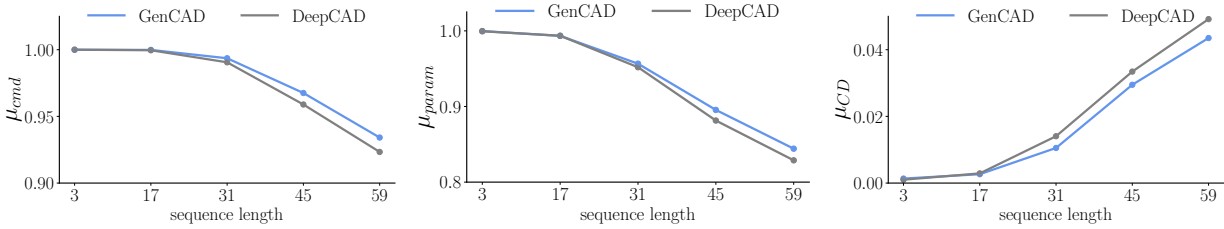

Figure 5: Performance of GenCAD-autoencoder corresponding to the length of CAD command sequences

Most importantly, our model is more accurate for higher lengths of CAD command sequences as shown in Figure 5. We anticipate that the high accuracy for longer sequence lengths is due to the autoregressive encoding of the latent space. By utilizing autoregressive training, we allow the model to encode more information about the CAD command sequence. Some qualitative examples of our model are shown in Figure 6. Our model can reconstruct the CAD programs more accurately for relatively complex shapes when compared to DeepCAD.

### 5.2 Image-based CAD program retrieval

While there are methods for image-based CAD-retrievals for specific datasets such as ROCA Gümeli et al. (2022) and DiffCAD (Gao et al., 2024), there are no existing image-based CAD program retrieval methods. For this reason, we compare our model against an image-to-image similarity metric. Using a pre-trained ResNet-18 architecture, we extract the image latents and find the cosine similarity between the ground

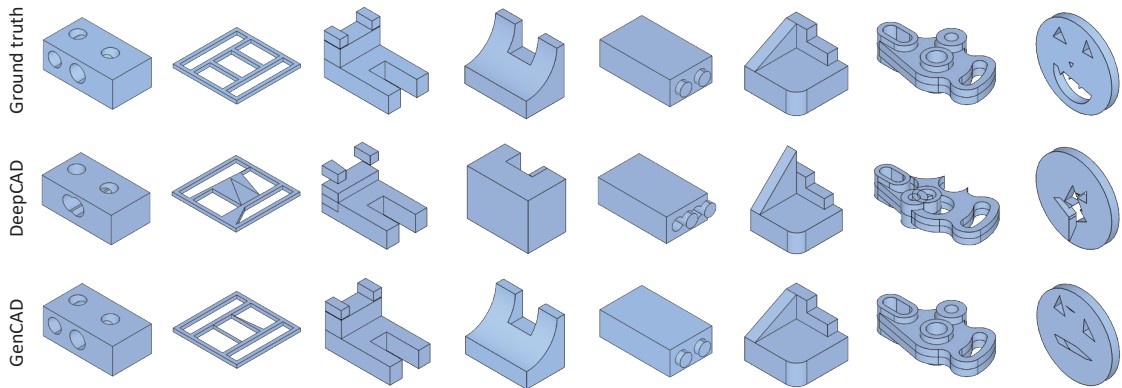

Figure 6: **3D CAD shape encoding performance:** We show the qualitative examples against DeepCAD which also uses a transformer-based model for CAD command reconstruction but does not use autoregressive training. Our method gives more accurate CAD command prediction for a test dataset that is close to the ground truth.

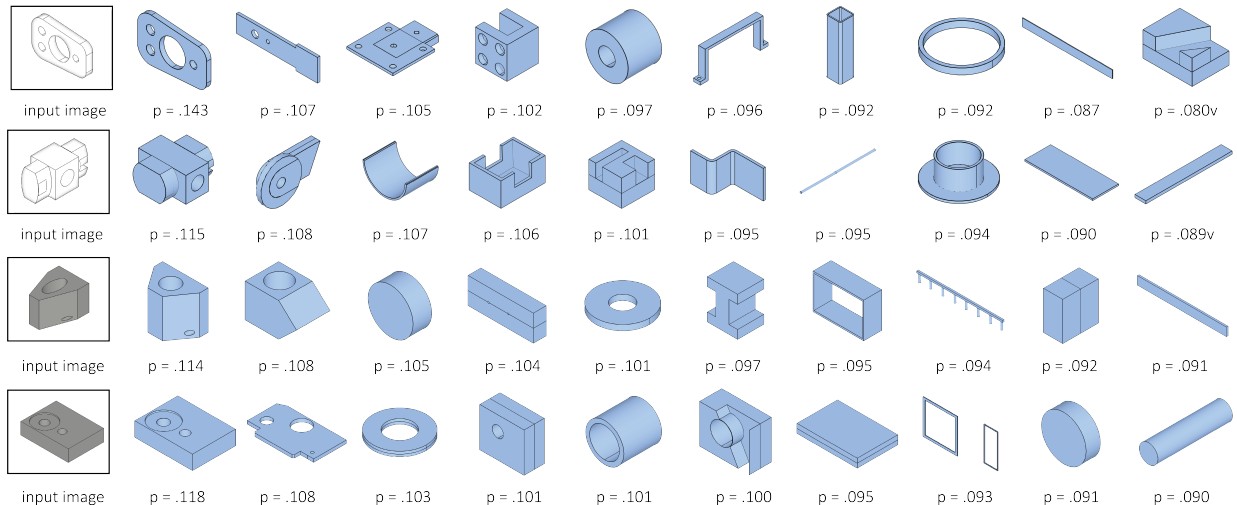

Figure 7: **Image-based CAD retrieval:** Qualitative performance of GenCAD-CCIP model is shown in terms of image-based CAD retrieval task for randomly sampled batches of 10 examples from the test dataset. The input CAD-image or CAD-sketch is shown on the left most column and the retrieved CAD programs are shown in the rest of the columns. We also provide the probability assigned by the CCIP model for each retrieval. The multimodal latent space helps in the accurate retrieval of CAD programs from input CAD images by assigning higher probabilities to similar CAD programs and lower probabilities to the rest of the CAD programs.

truth and the randomly sampled CAD images. We also provide random guesses as a simple baseline for comparison. As shown in Table 2, our model outperforms other baselines by a large margin. Note that our model is most accurate for small batches of CAD programs, and the accuracy drops when the number of CAD programs increases. Our model is also almost 61% accurate in retrieving CAD programs based on provided input images from a relatively large collection of CAD programs, i.e., 2048. Overall, the CCIP model is more than 15× accurate in retrieving the correct CAD program when compared to image-to-image search. Some qualitative examples of retrieval results are shown in Figure 7 and Figure 8. As the CAD programs are randomly sampled within the batch, a diverse number of shapes are present which makes the retrieval challenging. It is easy to see that our model provides higher probability values to the correct CAD program based on the image input. Additionally, similar shapes are assigned relatively higher probabilities

Table 2: Image-based and sketch-based CAD retrieval performance of GenCAD. $n_b = 10, 128, 1024$ and 2048 are repeated $1000, 10, 3$ and 3 times respectively. Arrows indicate whether higher is better or lower is better. Bold numbers represent the overall best performance.

| Method | $R_{B=10}(\uparrow)$ | $R_{B=128}(\uparrow)$ | $R_{B=1024}(\uparrow)$ | $R_{B=2048}(\uparrow)$ |
|---|---|---|---|---|
| Random | $10.06 \pm 0.19$ | N/A | N/A | N/A |
| ResNet-18 (pre-trained) | $77.70 \pm 0.00$ | $19.26 \pm 0.00$ | $5.21 \pm 0.45$ | $3.91 \pm 0.64$ |
| GenCAD–image | $\mathbf{98.49 \pm 3.93}$ | $\mathbf{91.41 \pm 0.6}$ | $70.28 \pm 0.79$ | $\mathbf{60.77 \pm 0.99}$ |
| GenCAD–sketch | $98.36 \pm 4.03$ | $87.5 \pm 3.49$ | $\mathbf{70.67 \pm 1.03}$ | $60.77 \pm 0.75$ |

compared to shapes that are different. This is more clearly seen in Figure 8 where we retrieve the top 5 CAD programs from a collection of 2048 CAD programs based on the provided input image. Our model extracts shapes that are similar to the input image and also provides diversity within the shape.

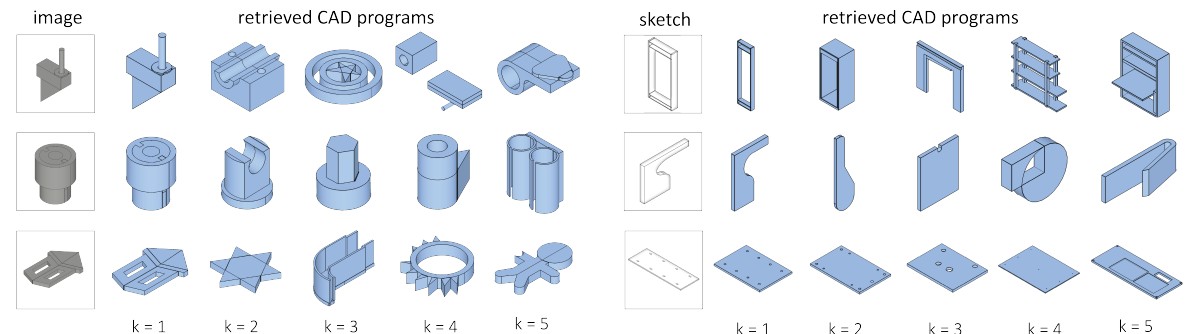

Figure 8: **Image-based and sketch-based CAD retrieval from large CAD database:** Qualitative performance of GenCAD-CCIP model is shown in terms of image-based retrieval of CAD programs from large database. Here, we retrieve top-k, where $k = 5$, CAD programs from the entire test dataset of CAD programs based on input images. The first column shows the input image or sketch and the rest of the columns show retrieved CAD based on the image.

## 5.3 Unconditional CAD program generation

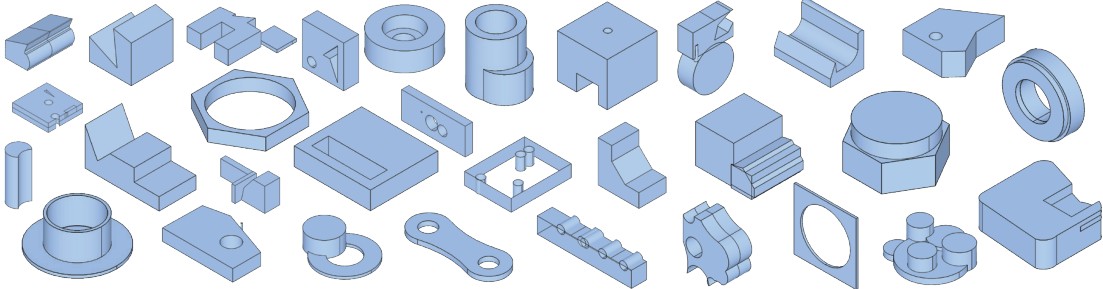

Figure 9: GenCAD (unconditional) 3D CAD generation.

As mentioned in section 4.3, we compare our unconditional CAD generation model against the DeepCAD latent GAN (l-GAN) model, the SkexGen model, the Brepgen model and the ContrastCAD model. Below we provide the performance in terms of coverage (COV), Maximum Mean Discrepancy (MMD), and Jensen Shannon Divergence (JSD). Our unconditional model outperforms both the DeepCAD l-GAN model and the SkexGen model in terms of both COV and MMD metrics as shown in Table 3. Our model also provides comparative performance to ContrastCAD and Brepgen. A few qualitative examples of generated 3D shapes

Table 3: 3D CAD (B-rep) shape generation performance. Arrows indicate whether higher is better or lower is better. Bold numbers represent the overall best performance.

| Method | type | COV (↑) | MMD (↓) | JSD (↓) |
|--------|------|---------|---------|---------|
| DeepCAD (Wu et al., 2021) | unconditional | 78.13 | 1.45 | 3.76 |
| SkexGen (Xu et al., 2022) | unconditional | 78.17 | 1.55 | 4.89 |
| Brepgen (Xu et al., 2024b) | unconditional | 73.10 | **1.05** | **1.22** |
| ContrastCAD + RRE (Jung et al., 2024) | unconditional | 78.93 | 1.44 | 3.67 |
| GenCAD | unconditional | 78.27 | 1.44 | 3.94 |
| GenCAD-image | conditional | **81.37** | 1.38 | 3.49 |
| GenCAD-sketch | conditional | **82.59** | 1.33 | 3.53 |

using the latent diffusion model are shown in Figure 9. Note that our model can generate complex CAD programs resulting in realistic 3D shapes.

## 5.4 Image conditional CAD program generation

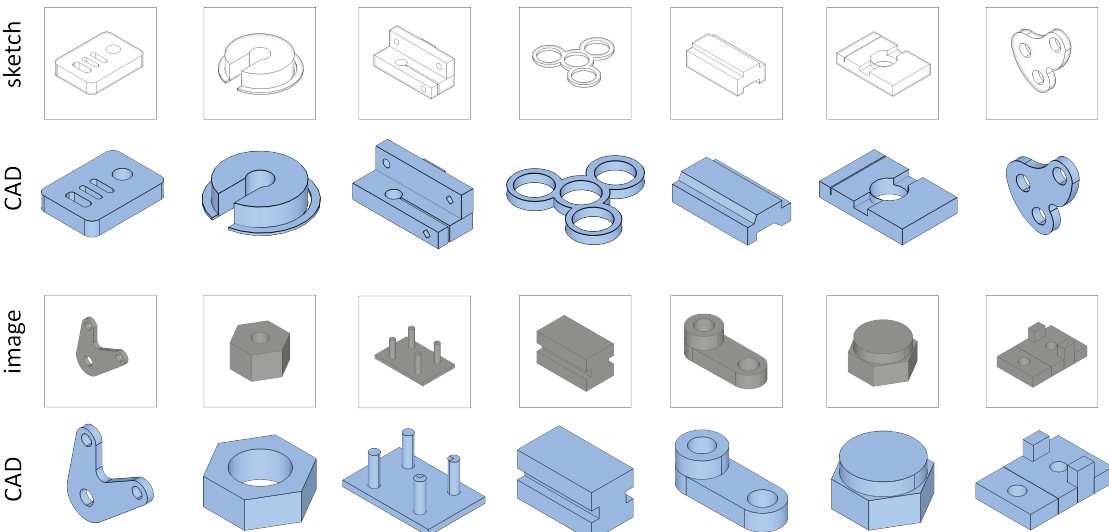

Figure 10: **GenCAD (Conditional) 3D CAD generation:** Qualitative examples of sketch-to-CAD and image-to-CAD generation capability of GenCAD using the conditional latent diffusion prior model.

In Table 3, we show that our conditional diffusion generative model outperforms DeepCAD l-GAN, SkexGen, ContrastCAD, and also our unconditional diffusion model in terms of COV, MMD, and JSD metrics. Note that we compare our model against the most capable version of ContrastCAD, which uses data augmentation. Additionally, our conditional model also significantly outperforms Brepgen in terms of the COV metric. Interestingly, Brepgen performs slightly better in the other two metrics. Note that COV represents the diversity of the generated shapes, while MMD represents the fidelity of the shapes and JSD represents the statistical distribution of the test set and the generated set. Thus, Brepgen performs better in terms of MMD and JSD by sacrificing the diversity of the shapes. We argue that GenCAD maintains a careful balance between diversity, fidelity, and statistical distance.

Finally, to quantify the alignment of the generated samples compared to the input images we compare the FID score between the distribution of CAD latents between a generated dataset and the test data set. As DeepCAD is the only available model that generates CAD program, we directly compare our results against DeepCAD model. As an additional baseline, we also compare our results against the deterministic-prior model from 4.3.3. Our results indicate that the image-conditional diffusion prior model provides the lowest

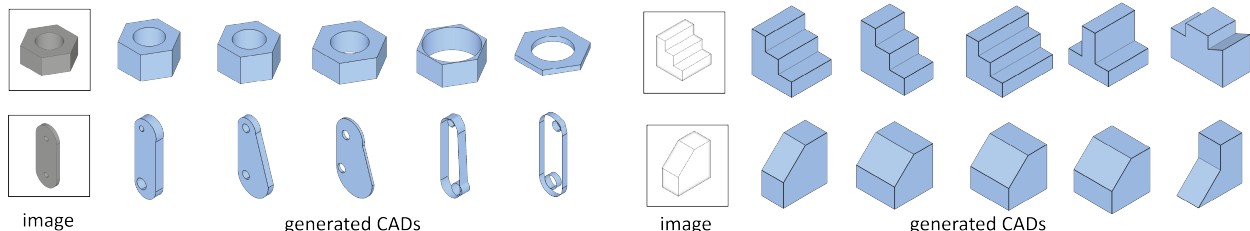

Figure 11: **Diversity of generated CAD:** Several samples, conditioned on image or sketch, drawn from the latent conditional model of GenCAD.

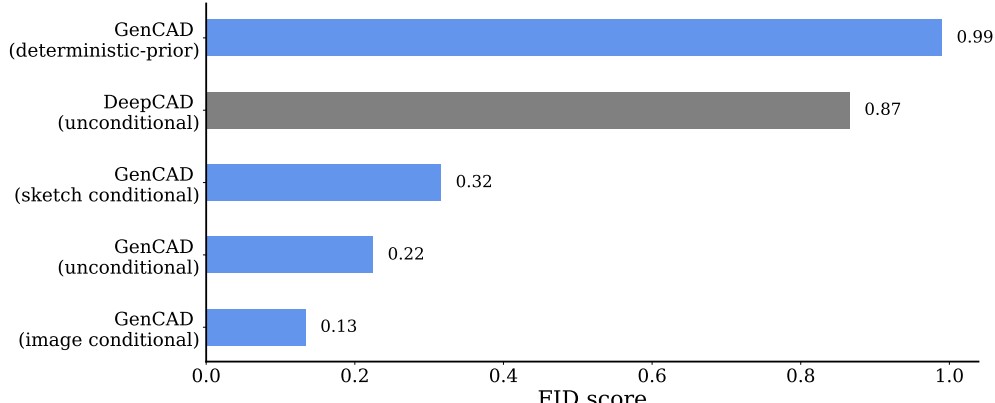

Figure 12: FID score of the generated dataset and the corresponding test dataset (lower is better).

FID score when compared to the sketch-conditional diffusion prior model, l-GAN, deterministic-prior, and unconditional diffusion model as shown in Figure 12. Note that the deterministic prior has the highest FID score and the unconditional diffusion model outperforms the l-GAN model. This result shows that the diffusion prior model can generate CAD programs that are close to the ground truth distribution. While the l-GAN model can generate unconditional CAD programs, these CAD models are not well aligned with the test dataset and the latent diffusion model is in-between the prior model and the l-GAN model. Finally, we anticipate that the FID score for the CAD-sketch model is higher due to its capability in generating more diverse shapes than the image-conditional model which is also supported by the COV metric in table 3. We also provide qualitative examples of our model in Figure 10.

Finally, we show the diversity of our diffusion prior model in Figure 11 where we sample multiple CAD programs for the same image input. Our model can generate similar CAD programs while providing useful variation in the parameter space.

# 6   LIMITATIONS AND FUTURE WORK

Although GenCAD shows impressive image-to-CAD performance, there are some limitations to the current version of this model. The CAD programs used in this study are comparatively simpler than industrial design tasks. The limited CAD vocabulary used in this study needs to be extended by including more sophisticated CAD tokens such as revolve operation, edge operation (e.g., fillets/chamfers), and other sketch operations. Additionally, GenCAD cannot guarantee the generation of a valid CAD similar to most of the generative CAD models reported in other studies. Drawing motivation from recent NLP literature, a step forward can be the addition of feedback from a CAD verification tool such as a geometry kernel. Building such a framework is non-trivial and will have the potential to create user-specified CAD programs. Finally, the images used in this study are mostly isometric and noise-free CAD-images. Future studies should focus on generating CAD programs from real-world object images with noisy backgrounds and non-isometric views.

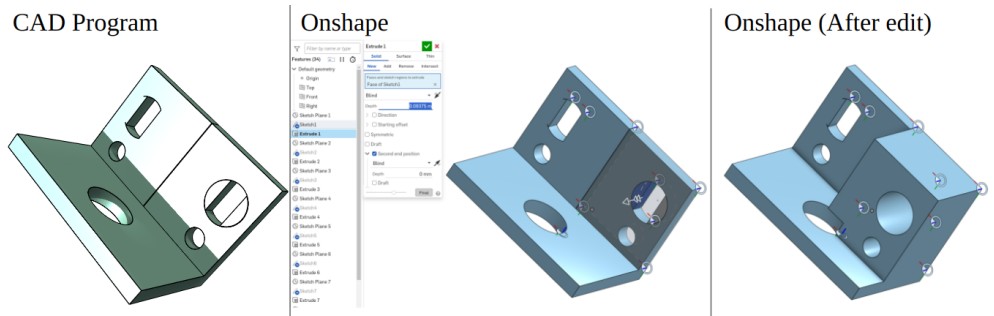

Figure 13: GenCAD generates the entire CAD program sequentially, which allows user-editability of the 3D model in commercial CAD software. Here we show a visualization of the original generated CAD program, the editable CAD in a commercial CAD software, e.g. Onshape, and the result of editing a feature.

Finally, to show the real-world applicability of GenCAD, we show its integration with a commercial CAD software ((Onshape, 2007)) in Figure 13 which enables user editability of the generated shape.

## 7 CONCLUSION

CAD program generation is a challenging task for AI models that have the potential to automate industrial design pipelines. While unconditional generation of CAD programs has been studied, a more important problem is to generate grounded CAD programs based on user intention. We provided GenCAD, the first generative model that can generate an entire CAD program for complex 3D shape conditioned on image or sketch inputs. We argue that the image-aligned CAD program generation has the potential to be directly utilized for design task acceleration. We also show the application of GenCAD in image-based CAD program retrieval tasks. In summary, GenCAD solves two long-standing challenges in the 3D CAD community: it can retrieve a CAD program from large databases using an image and also generate the CAD program based on the image. The current limitation of GenCAD is its dataset which includes relatively simpler CAD commands. In future studies, we aim to create a more sophisticated and real-world CAD dataset that can enhance the capabilities of GenCAD.

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
