# GenCAD: Image-Conditioned Computer-Aided Design Generation with Transformer-Based Contrastive Representation and Diffusion Priors

**Md Ferdous Alam**                                                                            *mfalam@mit.edu*
*Department of Mechanical Engineering*
*Massachusetts Institute of Technology*

**Faez Ahmed**                                                                                    *faez@mit.edu*
*Department of Mechanical Engineering*
*Massachusetts Institute of Technology*

**Reviewed on OpenReview:** *https://openreview.net/forum?id=e817c1wEZ6*

## References

Kaiming He, Xiangyu Zhang, Shaoqing Ren, and Jian Sun. Deep residual learning for image recognition. In *Proceedings of the IEEE conference on computer vision and pattern recognition*, pp. 770–778, 2016.

Rundi Wu, Chang Xiao, and Changxi Zheng. Deepcad: A deep generative network for computer-aided design models. In *Proceedings of the IEEE/CVF International Conference on Computer Vision*, pp. 6772–6782, 2021.

## A    CAD vocabulary

Here we provide the CAD vocabulary presented in this study where each token represents a CAD modeling operation or CAD command. This representation is based on (Wu et al., 2021). In total, there are three sketch tokens and one extrusion token. Note the CAD embedding, in this context, denotes the parameterized representation of each CAD modeling operation or token.

Table 1: A natural language equivalent representation of parameterized CAD commands using fixed-dimensional vectors. The blank square represents masked elements.

| index | token (CAD command, $t_i$) | embedding (parameters, $\mathbf{p}_i$) |
|:---:|:---:|:---:|
| 0 | $\langle \texttt{SOL} \rangle$ | $\emptyset$ |
| 1 | Line | $[x, y, \square, \square, \square, \square, \square, \square, \square, \square, \square, \square, \square, \square, \square, \square]$ |
| 2 | Arc | $[x, y, \alpha, f, \square, \square, \square, \square, \square, \square, \square, \square, \square, \square, \square, \square]$ |
| 3 | Circle | $[x, y, \square, \square, r, \square, \square, \square, \square, \square, \square, \square, \square, \square, \square, \square]$ |
| 4 | Extrude | $[\square, \square, \square, \square, \square, \theta, \phi, \gamma, p_x, p_y, p_z, s, e_1, e_2, b, u]$ |
| 5 | $\langle \texttt{EOS} \rangle$ | $\emptyset$ |

## B    GenCAD against image-to-mesh models

To show the capabilities of GenCAD, in terms of generating 3D shapes of mechanical objects, we compare GenCAD against a few state-of-the-art image-to-mesh models. While these models can create 3D mesh, the outputs are not accurate and not useful for engineering tasks. In contrast, GenCAD can create actual CAD model with intricate features which can be edited and modified by an expert engineer.

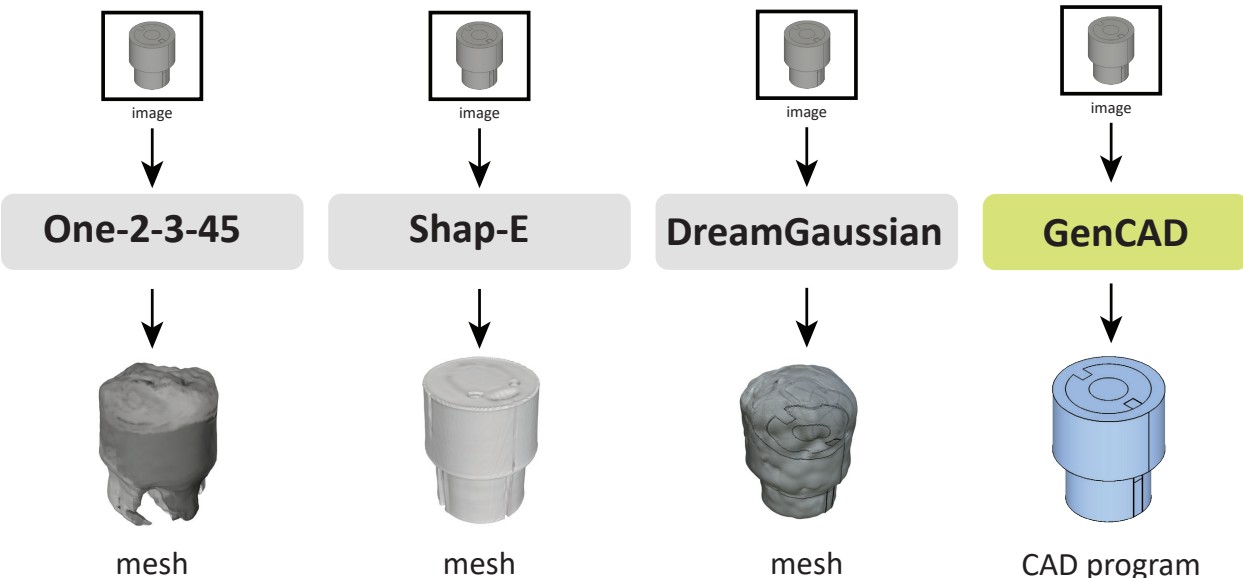

Figure 1: GenCAD vs image-to-mesh models: Here we show an example of GenCAD output against state-of-the-art large-scale image-to-mesh outputs. These models are not trained for solid modeling and, thus, fail to capture the intricate geometry features. In contrast, GenCAD can create the CAD and also provide the design history with accurate geometry features.

## C  Pseudocode

```
1  class GenCAD:
2      def __init__(self, csr, ccip, cdp):
3          self.csr = csr(...)   # frozen
4          self.ccip = ccip(...)   # frozen
5          self.cdp = cdp(...)   # frozen
6
7      def sample(self, condition):
8          ...
9
10
11  # CAD Sequence Reconstruction
12  csr = CADTransformer(encoder=..., decoder=...)
13  # Constrastive CAD-Image
14  ccip = CCIPModel(cad_model=csr.encoder)
15  # Latent conditional diffusion CAD
16  cdp = DiffusionPrior(ccip_model=ccip)
17  # GenCAD
18  gencad = GenCAD(csr=csr, ccip=ccip, cdp=cdp)
19
20  images = ...    # input images
21  cads = gencad.sample(condition=images)
```

## D  Ablation studies

We show ablation studies for the CSR model and the CCIP model. As the diffusion prior depends on both the CSR encoder and the CCIP image encoder, we do not ablate the diffusion prior model. In particular,

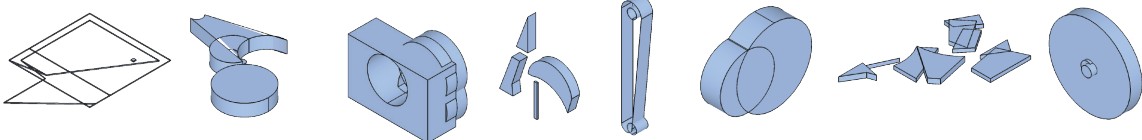

Figure 2: Limitations of GenCAD: Here we show a few of the failed samples generated by GenCAD. Note that intersecting faces, irregular shapes, and disjoint shapes are found in the generated shapes. We anticipate that most of these failed samples can be fixed by an expert designer.

we varied the number of encoder and decoder layers in the CSR model. Our results indicate that the 4-layer architec performs best in terms of command and parameter accuracy. The 2-layer architecture is slight better in terms of chamfer distance and invalid ratio. Eventually we pick the 4-layer architecture as the CSR model due to its superior capability in command and parameter reconstruction efficiency.

Table 2: Shape encoding performance of the proposed autoregressive encoder-decoder model. Arrows indicate whether higher is better or lower is better. Bold numbers represent the overall best performance.

| number of layers | $\mu_{\text{cmd}}(\uparrow)$ | $\mu_{\text{param}}(\uparrow)$ | $\mu_{\text{CD}}(\downarrow)$ | IR($\downarrow$) |
|---|---|---|---|---|
| $n_{\text{enc}} = 2, n_{\text{dec}} = 2$ | 99.43 | 97.50 | **0.754** | **2.03** |
| $n_{\text{enc}} = 4, n_{\text{dec}} = 4$ | **99.51** | **97.78** | 0.762 | 3.32 |
| $n_{\text{enc}} = 6, n_{\text{dec}} = 6$ | 94.25 | 95.20 | 3.13 | 14.94 |

For the CCIP model, we experiment with different resnet architectures and the ViT model as the sketch encoder model. For the ViT model, we use 6 attention layers with 16 heads and a patch size of 32. Our results show that the ResNet-18 sketch encoder model outperforms both the smallest model, ResNet-10, the largest model, ResNet-34, and the ViT model. As a result, we use the ResNet-18 as the sketch encoder for the CCIP model.

# E   Architecture and training details

## E.1   CSR model

The encoder part of the CSR model consists of 4 transformer encoder layers with 8 attention heads. The decoder of the CSR model consists of 4 transformer decoder layer with 8 attention heads. Both the encoder and decoder use a 512 dimensional feed-forward network in the attention layer with a dropout value of 0.1. The CSR model is trained using Adam optimizer with a batch size of 512. The initial learning rate is $1e-3$ with GradualWarmupScheduler for 2000 warmup steps. The model is trained for 1000 epochs. Additionally, the gradient is clipped at 1.0 during training.

Table 3: Image-encoder ablation. $n_b = 10, 128, 1024$ and 2048 are repeated 1000, 10, 3 and 3 times respectively. Arrows indicate whether higher is better or lower is better. Bold numbers represent the overall best performance.

| GenCAD image encoder | $R_{B=10}(\uparrow)$ | $R_{B=128}(\uparrow)$ | $R_{B=1024}(\uparrow)$ | $R_{B=2048}(\uparrow)$ |
|---|---|---|---|---|
| ResNet-10 | $97.31 \pm 5.28$ | $85 \pm 2.72$ | $57.74 \pm 0.7$ | $49.52 \pm 0.78$ |
| ResNet-18 | $\mathbf{98.49 \pm 3.93}$ | $\mathbf{91.41 \pm 0.6}$ | $\mathbf{70.28 \pm 0.79}$ | $\mathbf{60.77 \pm 0.99}$ |
| ResNet-34 | $98.14 \pm 4.37$ | $85.55 \pm 3.53$ | $64.25 \pm 0.84$ | $52.12 \pm 0.66$ |
| ViT | $96.62 \pm 5.81$ | $85.23 \pm 3.47$ | $62.43 \pm 1.05$ | $53.87 \pm 0.52$ |

### E.2 CCIP model

The CCIP model consists of the pre-trained encoder model of the CSR as the CAD encoder and a ResNet-18 architecture for the image encoder. The ResNet-18 architecture is similar to He et al. (2016) with an additional dropout layer between the first batch normalization layer and the ReLU activation layer of each block. For the image encoder, we use a dropout value of 0.1. The model is trained using Adam optimizer with weight decay. An initial learning rate of $1e-3$ is used with a ReduceLROnPlateau scheduler. The model is trained for 500 epoch with a batch size of 256.

### E.3 CDP model

The LDM model uses 500 timesteps for the diffusion process. The model uses 10 blocks of MLP-resnet network with a 2048 dimensional projection layer and 0.1 dropout. The gradient is accumulated every 2 steps during training. We use a fixed learning rate of $1e-5$ with a batch size of 2048. The model is trained for $1e6$ timesteps.

### E.4 Evaluation metrics

**Coverage (COV):** We use COV to measure the diversity of the generated shapes. To quantify the diversity, the fraction of shapes in the reference set $\mathcal{S}$ that are matched by at least one shape in the generated set $\mathcal{G}$. Formally, COV can be defined as:

$$COV(\mathcal{S},\mathcal{G}) = \frac{\{argmin_{Y\in\mathcal{S}}d_{CD}(X,Y)|X\in\mathcal{G}\}}{|\mathcal{S}|},$$

where $d_{CD}$ is the chamfer distance between two point clouds $X$ and $Y$. We uniformly sample 2000 points in each of the shapes and use a set of 7000 generated shapes for each model.

**Minimum matching distance (MMD):** We use MMD to calculate the fidelity of the generated shapes. For each shape in the reference set $\mathcal{G}$, the chamfer distance to its nearest neighbor in the generated set $\mathcal{G}$ is computed. MMD is defined as the average over all the nearest distances:

$$MMD(\mathcal{S},\mathcal{G}) = \frac{1}{|\mathcal{S}|}\sum_{Y\in\mathcal{S}}\min_{X\in\mathcal{G}}d_{CD}(X,Y).$$

**Jensen-Shannon Divergence (JSD):** We use JSD to calculate the statistical distance between the distribution of generated shapes in $\mathcal{G}$ and the distribution of reference shapes in $\mathcal{S}$. To calculate JSD we use the following formula using KL-divergence:

$$JSD(\mathcal{P}_{\mathcal{S}},\mathcal{P}_{\mathcal{G}}) = \frac{1}{2}D_{KL}(\mathcal{P}_{\mathcal{S}}||M) + \frac{1}{2}D_{KL}(\mathcal{P}_{\mathcal{G}}||M)$$

where $M = \frac{\mathcal{P}_{\mathcal{S}}+\mathcal{P}_{\mathcal{G}}}{2}$ and $\mathcal{P}_{\mathcal{G}}$ and $\mathcal{P}_{\mathcal{S}}$ are the marginal distribution of points of the generated and reference sets, respectively. To directly compare our results against other benchmarks, we follow the same calculation procedure as reported in literature. A set of 1000 randomly sampled shapes are used to construct the reference set and 3000 generated shapes are used as the generated set.