# OpenReview forum: "GenCAD: Image-Conditioned Computer-Aided Design Generation with Transformer-Based Contrastive Representation and Diffusion Priors"
_TMLR — Accepted by TMLR_

### Review · Reviewer_5nfB · 2024-09-17

**Summary Of Contributions:**

They propose a solution called GenCAD to generate/retrieve CAD programs/models from image inputs.

They first use a transformer-based encoder-decoder architecture to auto-regressively learn the latent features for generating CAD programs that represent the CAD model in B-rep. They then utilize contrastive learning to learn a joint latent space between CADs and images. Additionally, they train a diffusion model to generate CAD latents from image inputs. Finally, CAD programs can be decoded using the transformer's decoder.

I believe their main focus is on enhancing a previous work, DeepCAD, which converts CAD into a language model. The authors' enhancements include:
1. Conditional generation from images.
2. Improvements in diversity, fidelity, and accuracy.

As for the technical contributions they claimed in the paper (see the end of Section 1):
1. A transformer-based autoregressive model: SolidGen also proposed using an autoregressive transformer for conditioned/unconditioned generation. Therefore, I find this point less convincing.
2. A conditional CAD generative model from image inputs: This contribution is not particularly novel as SolidGen has already implemented it.
3. Contrastive learning: This contribution appears trivial, as ContrastCAD has undertaken a similar approach.

**Audience:**

Yes

**Claims And Evidence:**

Yes

**Requested Changes:**

See the above weak points. Requested changes:
1. More comparisons are needed in retrieval experiments, *e.g.*, DiffCAD and ROCA.
2. Compare it against SolidGen and ContrastCAD.
3. Revise any typos and inappropriate wordings.

**Strengths And Weaknesses:**

Strengths they claimed:
- They enable both conditional and unconditional generation of CAD models.
- The performance is clearly superior to that of DeepCAD and SkexGen.


Weaknesses:
1. Incremental contributions; novelty is not enough; the experiments are not yet convincing.
2. Retrieval experiments are incomplete.
   - The authors stated, "as there are no existing image-based CAD retrieval methods, we compare our model against an image-to-image similarity metric." This assertion is incorrect. Numerous image-based CAD retrieval methods exist. For instance, DiffCAD and ROCA can even handle CAD retrieval using real image inputs. It would be more appropriate to compare and conduct experiments on both synthetic and real image inputs.
   - References:
     - DiffCAD: Weakly-Supervised Probabilistic CAD Model Retrieval and Alignment from an RGB Image
     - ROCA: Robust CAD Model Retrieval and Alignment from a Single Image

3. SolidGen performed similar tasks; however, it does not generate CAD programs. It would be valuable to see whether producing CAD programs is clearly more effective and better than direct B-rep synthesis. Your paper did not present any evidence.

4. Some claims are perhaps overstated. I would recommend that the authors carefully revise their wording and include more comparisons and discussions in the related work section. For example:
   - In Section 7, the authors stated, "We provided a first-of-its-kind AI model, GenCAD, that can generate CAD programs for complex 3D shapes both unconditionally and also based on image inputs." However, the term "first-of-its-kind" is likely overstated, as another work called "SolidGen" also supports both unconditioned generation and conditioned generation from image inputs. This work was published in TMLR-2023.

5. Some experimental setups are not very practical. In Sections 5.2 and 5.4, how can we acquire such an input image in a practical scenario? I believe it would be challenging to use this in a real-world situation, as images captured by a camera do not resemble those shown.

6. Please consider comparing the latest methods on CAD generation in the appropriate experimental sections, including but not limited to:
   - "SolidGen: An Autoregressive Model for Direct B-rep Synthesis": This paper proposes using an auto-regressive transformer to generate CADs. It also supports unconditioned generation and image-conditioned generation.
   - "ContrastCAD: Contrastive Learning-based Representation Learning for Computer-Aided Design Models": It also utilizes contrastive learning techniques to effectively capture information and can handle long construction sequences as well.

7. They mentioned in Section 2.1 that DeepCAD has a limited number of sketches and 3D operations; however, GenCAD does not address this point, and I am unsure why they mention this limitation.

8. Typos (please revise your paper carefully again):
   - The second paragraph of Section 3.1:
     - The notation $\mathbf{R}^{17}$ should be changed to $\mathbb{R}^{17}$
     - The notation $\mathbf{t}_i \in \mathbb{R}$ should be changed to $t_i \in \mathbb{R}$
   - Section 5.2:
     - You typed "image-tp-image"

---

> ### Author Response · Authors · 2025-01-12
> **Response to reviewer 5nfB**
>
> We thank the reviewer for providing thorough and detailed comments. These comments will help us improve our work. We provide our responses to specific comments in the following:
>
> 1. Our experiments show significant improvement over existing state-of-art models. We also added new results, showing how the model generalizes to new types of inputs (sketches). We also consider the proposed conditional architecture novel for CAD program synthesis as we have not found any prior work on image-to-CAD-program. Note that we create the entire sequential CAD modeling operations and not the final B-rep model. We would also highlight our architecture on training a contrastive model to learn multimodal data representations for CAD sequences is also unique as it allows joint embeddings of multiple modalities.
>
> 2. We would like to mention that the retrieval experiments report the retrieval of sequential CAD programs instead of only the 3D model. ROCA and DiffCAD are developed for specific datasets such as scan2CAD which does not contain general mechanical shapes such as the dataset used in our paper. This difference in dataset makes it difficult to compare our model with ROCA or DiffCAD. To improve the clarity of the paper, we have taken the suggestion of the reviewer and reworded the sentence to mention that we retrieved the sequential CAD program instead of the CAD model and cited other efforts in retrieval tasks for different representations.
>
> 3. Thank you for pointing out the need for a clear distinction between CAD sequence generation and B-rep synthesis. To illustrate this fundamental difference, we have modified Fig. 1 and added a new visualization of our sequence imported into a commercial CAD software (e.g. Onshape) in the revised manuscript (see Fig. 13). This visualization clearly demonstrates that a feature tree from a CAD program allows direct editing of any step in the sequence, offering a dynamic, modifiable structure. In contrast, B-rep provides only a static model. Our comparisons with B-rep generators in the paper aim to demonstrate that, despite the primary focus on sequence generation, our model can compete with, and even outperform, some B-rep generators on standard CAD metrics. However, it's crucial to reiterate that the task of generating a modifiable CAD sequence is inherently different and our primary objective.
>
> 4. Thank you for your feedback. We have revised Section 7 to more accurately reflect the scope of our contribution. While SolidGen also supports both unconditioned and image-conditioned generation, it primarily produces B-rep models. In contrast, GenCAD uniquely generates entire sequential CAD programs, allowing dynamic editing of design steps. This capability sets GenCAD apart, emphasizing the practical applications of editable CAD programming, a novel approach in the field. We have clarified these distinctions in the revised manuscript.
>
> 5. Thank you for your observations regarding the practicality of our experimental setups. The primary aim of our image inputs is to facilitate reverse engineering of CAD models. Platforms like GrabCAD offer extensive CAD model repositories but lack associated design histories, which our method can sequentially reconstruct from image inputs. Additionally, our approach can generate CAD sequences from images of B-rep models, such as those created by SolidGen, thus enhancing existing models with editable sequences. Recognizing your concern about the realism of input images, we have expanded our experiments to include pencil-sketch inputs, demonstrating GenCAD's ability to generate complete CAD programs from more accessible and realistic inputs. This flexibility underscores the broad applicability of our framework beyond the specific examples shown in the initial manuscript.
>
> 6. We thank the reviewer for providing these references. Based on this feedback, we have compared our results with ContrastCAD. Note that ContrastCAD only handles unconditional generation. Our unconditional model performs closely to ContrastCAD while our conditional model outperforms ContrastCAD by a significant margin. Unfortunately, the SolidGen models are not publicly available and thus we are unable to compare this to our model. We contacted the authors but did not receive any response about their trained models. In addition to this, we have compared our model against another B-rep generation model, Brepgen [1], in Table 3.
>
> 7. We address this point as a limitation of our work which we plan to improve in future studies. Based on the feedback, we have reworded the sentence and provided a separate limitation section.
>
> 8. Thank you for noticing the issues. We have revised the paper and fixed the typos.
>
> [1] Xu, Xiang, et al. "BrepGen: A B-rep Generative Diffusion Model with Structured Latent Geometry." arXiv preprint arXiv:2401.15563 (2024).

---

### Review · Reviewer_EWNT · 2024-10-22

**Summary Of Contributions:**

# Summary
This paper introduces the GenCAD generative model, which utilizes an autoregressive transformer with a contrastive learning framework and a latent diffusion model to convert image inputs into parametric CAD command sequences, resulting in editable 3D shape representations. Results show that GenCAD outperforms existing state-of-the-art methods in generation of CAD models. Additionally, GenCAD's contrastive learning framework notably enhances the retrieval of CAD models through image queries.

**Audience:**

Yes

**Broader Impact Concerns:**

None,

**Claims And Evidence:**

Yes

**Requested Changes:**

see weaknesses parts.

**Strengths And Weaknesses:**

## Strengths:
- Aligning the latent space of image and CAD model enable accurate retrieval compared with image-to-image search.
- It appears to work better for obvious reasons. (Obvious reasons are fine; it makes sense.)
##  Weaknesses:
- The use of latent representations for CAD programs may introduces a significant limitation. Specifically, I think this approach may not scale effectively to more complex CAD models or larger dataset(compared with Autoregresive models.). It would be beneficial for the authors to discuss this limitation and its potential impact on the applicability of the method in the paper.
- lacking necessary Ablation Studies to prove the design of model(Crucial).

---

> ### Author Response · Authors · 2025-01-12
> **Response to reviewer EWNT**
>
> We thank the reviewer for providing insightful feedback and helping us improve the paper. We address the specific comments in the following:
>
> 1. We thank the reviewer for this important point. We agree that latent representation learning of CAD programs is a difficult task, however, this approach allows us to create latent generative models for scalable generation of CAD sequences. Note that our framework uses an autoregressively trained decoder to generate the CAD sequence. As a result, our framework retains the scalability of autoregressive models while focusing on a CAD-specific language. Our quantitative evaluations show that latent sequence representation outperforms direct autoregressive generation of B-reps (as shown in Table. 3). In addition to this, the latent representation of CAD sequences enables multi-modality in the GenCAD framework. We agree with the reviewer that latent representation based generation should be investigated for large CAD sequences. To address this point, we identify two areas of exploration for future studies: 1) Augmenting the CAD sequence dataset with large sequence length and 2) Augmenting the CAD language representation.
>
> 2. We thank the reviewer for addressing this important issue. In the revised draft, we have added ablation studies of the transformer autoencoder and image encoder model. Specifically, we have provided the following in the supplement material: 1) Table 2 shows varied number of encoder/decoder layers in the transformer autoencoder, 2) Table 3 shows varied architecture for the image encoder. More specifically, we have changed the number of encoder and decoder layers in the CSR model and the resnet model size in the CCIP model. As the latent diffusion model depends on a pre-trained cad encoder and an image encoder, we utilized the best-performing models from these ablation studies to train the latent diffusion model. In addition to the above, we have added new experiments to show the capabilities of the GenCAD framework. While keeping the rest of the framework unchanged, we trained the image encoder with sketches of CAD shapes and trained the corresponding diffusion model as well. Our results, as shown in Table 3., show similar performance for both image inputs and sketch inputs.

---

### Review · Reviewer_YRo2 · 2024-12-18

**Summary Of Contributions:**

This paper propose a framework, dubbed GenCAD, for generating CAD models given a single image, which utlize contrastive representation and diffusion model, therefore enabling single-image-to-CAD and CAD model retrieval. GenCAD contains the following contributions:
1. a transformer-based model for CAD sequences encoding and reconstruction
2. an image-conditioned diffusion model for CAD generation
3. contranstive learning between image and CAD, which facilitate fast and accurate CAD retrieval

**Audience:**

Yes

**Broader Impact Concerns:**

No ethical concerns here.

**Claims And Evidence:**

No

**Requested Changes:**

1. As illustrated in weakness 1, provide more complex object generation results.
2. As illustrated in weakness 2, add the comparison experiment with  (variational) auto-encoder and correct the figure 2.
3. As illustrated in weakness 3, add comparison experiment with DiT.

**Strengths And Weaknesses:**

Strength:
1. GenCAD propose the first image-conditioned CAD generation model. Compared to its previous unconditional version, DeepCAD, it enables more flexible CAD generation that can be controled by users
2. GenCAD use contrastive leaning between images and CAD models, which facilitates CAD model retrieval

Weakness:
1. compared to current image-to-3d model, which use NeRF, gaussian splatting or mesh to represent the shape, CAD lacks capability to represent complex objects that have complicate topology and textures. The results shown in the paper only contains simple workpieces, it would be better to show its effectiveness on more complex furniture datasets, like [1].
2. The architecture of CSR model is confusing. CSR model is an auto-encoder model to encode and reconstruct the input CAD model, while autoregressive model is a generation model for generating the next token given previous ones. The standard operation of CSR model is to encode CAD sequence from c_1 to c_N and then decode it from c_1 to c_N, while the current CSR is to decode it from c_2 to c_N+1. Furthremore, because transformer is an N-in-N-out architecture, it's also confusing of using trapezoid rather than rectangular shape to represent encoder and decoder in Figure 2. The author should illustrate the motivation and advantage of such design in CSR model and provide the comparison between standard (variational) auto-encoder model with current model.
3. The architecture design for CDP is also wired. Compared to current design, it would be better to use DiT architecture, which use transformer architecture and regard condition as a injection latent injected in the model via cross attention. It would be better to provide comparison experiment between current design and DiT.


[1] BrepGen: A B-rep Generative Diffusion Model with Structured Latent Geometry
[2] Scalable Diffusion Models with Transformers

---

> ### Author Response · Authors · 2025-01-12
> **Response to reviewer YRo2**
>
> We thank the reviewer for providing important feedback and would like to address specific comments in the following:
>
> 1. We appreciate the reviewer's comments and recognize the concern about the complexity of objects represented in our results. It's important to clarify that GenCAD's core contribution is the generation of CAD programs, which inherently differ from 3D model representations like meshes or volumes used in NeRF or similar techniques. While these methods capture complex topologies and textures effectively, CAD's strengths lie in its editability and its industry-standard status for detailed, manufacturable design in engineering. This distinction makes direct comparisons challenging, as each modality serves different end purposes. CAD's utility in real-world applications such as reverse engineering, customization, and manufacturability cannot be overstated. Our objective with GenCAD is not to argue which representation is superior but to showcase a novel method for generating CAD programs, which has significant practical implications, especially in engineering and design industries where modifications to designs are frequent and critical. GenCAD demonstrates an example of generating the entire CAD program sequentially from an image input. Note that, unlike existing conditional generative CAD models, GenCAD creates the entire CAD program instead of the final design. This provides the additional benefit that a user can edit the model at any step by directly importing it into their CAD software (see Fig. 13). Thank you for highlighting the furniture dataset. Our study currently does not include this dataset due to its inclusion of CAD operations that exceed our model's existing vocabulary, as detailed in Table 1 in Supplement A. The vocabulary used by us is commonly used in the AI for CAD literature. While enriching this vocabulary will undoubtedly enhance our model's capability to represent more complex CAD models, our present focus is on establishing a robust methodology for generating CAD programs within a defined vocabulary and we show that we obtain state-of-art results compared to existing AI models. Consequently, incorporating the furniture dataset falls outside the scope of this paper. However, it's important to note that once future work expands the CAD vocabulary to include additional operations such as mirroring, sweeping, filleting, and chamfering, our approach can be readily adapted to apply these more complex operations, thus broadening the potential applications of our work. Finally, to address your comment, we have compared our model against BrepGen, and our model outperforms BrepGen in terms of diverse shape generation (as shown in Table 3).
>
> 2.  We would like to clarify that our CSR model is an autoregressive transformer model. The $c_1$ token is the start-of-the-loop token or $<SOL>$ token in (Table 1 in Supplement A). We do not predict this token while decoding. Instead, the decoder takes the latent vector and autoregressively predicts the next token from the previous ones. For example, for an input sequence $(<SOL>, Line, Line, Arc, Circle, Extrude, <EOS>)$, the encoder receives this sequence and generates the latent vector. The decoder then receives the latent vector as input and generates the following sequence as the decoded output $(Line, Line, Arc, Circle, Extrude, <EOS>)$. For this reason, we use the notations mentioned by the reviewer. We would like to thank the reviewer for addressing the visualization in Figure 2. Based on the reviewer's feedback we have recreated the figure with transformers as rectangular shapes instead of trapezoidal shape. We utilize transformer based model because of its capability to take sequential input and train the diffusion model separately to make our framework scalable. Because each part of our model is trained separately, it is scalable and can be trained with a large amount of data. As the goal of the CSR model is not to work as a generative model for CAD sequences but to only reconstruct it for learning latent representation, a variational autoencoder is not useful in this case. For completeness, we compare our model against similar standard generative models such as latent GAN model from DeepCAD in Table 3.
>
> 3. Our CDP model has a ResNet-MLP architecture as the denoising model (figure 4). This design choice is based on our experiments. We would also like to mention that the encoder of our CSR model creates a sequence of latent vectors which are then average-pooled to create one $256$-dimensional latent vector. The CSR decoder receives this single latent vector as input (Figure 2). As our latent representation is a $1\times 256$ dimensional vector instead of a sequence of latent vectors, it is not possible to use Diffusion Transformers type models in this case without changing the proposed architecture. Note that our choice of average encoder output is based on our experiments and accuracy in creating the CAD sequence.

---

### Author Response · Authors · 2025-01-12
**Summary response**

We sincerely thank the reviewers for their thorough and constructive feedback. We appreciate the time and effort they have dedicated to evaluating our work, and their insights will greatly contribute to improving the quality and clarity of our paper. We have provided individual comments to address each reviewer's concerns. In the following, we provide the major changes in the revised paper.

1. To improve the clarity of our work and distinguish it from direct B-rep synthesis, we updated figure 1 and showed the sequential generation of CAD.
2. We have added one additional figure that shows the user-editability of the model output in a commercial CAD software
3. To show the capability of our model, we have trained our architecture on a new dataset of synthetic CAD-sketches and reported the results in table 2 and table 3.
4. We have also updated figures 7, 8, 10, 11 with qualitative outputs from new experiments
5. We have compared our model with additional two baselines, namely ContrastCAD and Brepggen in table 3.
6. Ablation studies for our model has been added in table 2 and table 3 in the supplement material

---

### Public Comment · ~Milin_Kodnongbua1 · 2025-01-17
**Technical results**

I enjoyed reading the paper, this works present a novel approach in combining CAD command generation and image modalities. I have read the reviews, author comments, and the revision. However, I have a few concerns about the validity of the results:

1. For unconditional generation, there is no analysis of the novelty of the generated shapes (whether the model memorizes the training data or whether it generates something new). In the revision, the authors added the COV, MMD, and JSD metrics. However, the descriptions of the metrics are vague and cannot be reproduced. In other works, COV metric usually represents the coverage of the generated data on the training data. I was wondering why the COV metrics the authors reported were different that what were reported in BrepGen, but I am assuming the authors are using a different "diversity" COV metric.
2. For image conditional generation, how are the authors pick the input images? Are they renderings of the models in the testing DeepCAD dataset?
3. In figure 10, how are the authors pick the generated CAD models to display when the model can generate a variety of CAD models (in Figure 11)?
4. How do the authors compute the COV metric for the image-conditioned experiments? Why do they perform better in terms of "diversity"? Are the metrics for using one input image or multiple images as conditioning?

---

> ### Author Response · Authors · 2025-01-17
> **response to your comment**
>
> Thank you for reading our work and providing comments. We are happy to know that you enjoyed our paper and glad to address your comments.
>
> 1. We use the COV metric to show the diversity of the samples in terms of coverage of generated shapes on the training data. Note that we use the standard metrics that have been reported in the literature [1, 2, 3, 4]. As our goal is to demonstrate the capabilities of the conditional model and have the same comparison metrics for all models, we did not provide any specific novelty results for the unconditional model.  We would like to note that we followed the same COV, JSD and MMD metrics from DeepCAD [1] as it has been fairly standard in other generative CAD works in recent literature. There is variation in reported values for these metrics for the baseline DeepCAD model in literature [1, 2, 3, 4]. To overcome this issue and have a fair comparison across all metrics, we generated a set of $\approx 7500$ shapes to calculate all metrics and kept the rest of the experimental setting same as DeepCAD. The value of COV for BrepGen is slightly different in our work as we calculated the metric by running the publicly available codebase for BrepGen. We did not directly report the value from their paper as the reported values for DeepCAD is different in Brepgen from the original DeepCAD paper. Our results are standardized on the DeepCAD as the baseline due to its direct similarity in CAD sequence generation. Additional note: the COV, MMD, JSD metrics are not new in the revision and have been added in the original draft as well. We added new baselines in the revision. The description of the metrics are provided in the supplement materials in details.
>
> 2. Yes, the images are obtained by rendering the DeepCAD dataset. We then use the same train, validation, test split of the images to train, validate and test our model. For the diffusion prior, we use the isometric image of the rendering as the condition.
>
> 3. Figure 10 shows qualitative samples as output from our model. We generated one sample for each image and included that in figure 10. The purpose of figure 11 is to show that our model is capable of creating diverse samples for the same input image condition.
>
> 4. The COV metric calculation is similar to the unconditional model. First, we generate CAD for each test image which results in a set of approximately $\approx 7500$ shapes. Next we randomly select $3000$ shapes to calculate the COV metric. The experiment is repeated three times and the mean value is reported in the paper. There can be multiple reasons to examplain the diversity increase for the conditional generative model. For example, the conditioning helps the model to generate samples that are less similar to the training samples and hence, provides more diversity. We agree that this is an important question that should be explored in details in future studies. The metrics are calculated for one input image for this study.
>
> [1] Wu, Rundi, Chang Xiao, and Changxi Zheng. "Deepcad: A deep generative network for computer-aided design models." Proceedings of the IEEE/CVF International Conference on Computer Vision. 2021.
>
> [2] Jung, Minseop, Minseong Kim, and Jibum Kim. "ContrastCAD: Contrastive Learning-based Representation Learning for Computer-Aided Design Models." arXiv preprint arXiv:2404.01645 (2024).
>
> [3] Xu, Xiang, et al. "Skexgen: Autoregressive generation of cad construction sequences with disentangled codebooks." arXiv preprint arXiv:2207.04632 (2022).
>
> [4] Xu, Xiang, et al. "Brepgen: A b-rep generative diffusion model with structured latent geometry." ACM Transactions on Graphics (TOG) 43.4 (2024): 1-14.

---

### Decision · Action_Editor_pNb2 · 2025-02-19

**Recommendation:** Accept with minor revision

**Comment:**

This paper works on learning to generate CADs, including conditioning on sketch or image inputs. Technically, it proposes GenCAD that is based on autoregressive transformers with contrastive learning, and the parametric CAD command sequences are generated by using latent diffusion. Contrastive learning also enables the retrieval of CAD models using image queries from large CAD databases.

Reviewer 5nfB suggests to improve the completeness of retrieval experiments, present more fair comparisons with SolidGen and ContrastCAD, and make the experiment settings more practical. Reviewer EWNT pointed out that the paper is short of ablation studies and the proposed method may not deal with complex CADs. The authors have carefully responded to these comments, with necessary revisions that improve the paper quality greatly. Reviewer YRo2 initially raised concerns on architecture of CSR and how the proposed architecture design for CDP compares against DiT based one; for the former one, the authors have taken the suggestion and improved Fig. 2, and for the latter one, the authors explained why DiT cannot be directly applied to this context. AE has read the paper and partially agrees with the authors.

However, AE still suggests that the authors follow the reviewer's suggestion to compare with DiT, e.g., by NOT average-pooling the sequence of latent vectors generated by CSR encoder.

**Audience:**

Yes

**Claims And Evidence:**

Almost

---

> ### Author Response · Authors · 2025-03-06
> **Official comment by authors**
>
> We appreciate the reviewers' valuable feedback and the AE's efforts and decision. We will incorporate the minor revisions suggested by the AE and look forward to submitting the revised manuscript.